# Impact of crisis intervention on mental health in the context of specific civilian emergencies

**Xiaoshan Hu, Jihua Liu, Bingyu Hao, Yang Lv** [ID]*

College of Teachers, Chengdu University, Chengdu, China,

* lvyy0310@163.comti

## Abstract

### Background

The implementation of crisis response strategies, such as natural hazards, pandemics, and conflicts, is necessary during times of emergency. Despite the importance of these interventions, mental health outcomes in emergency situations remain poorly understood. There is a lack of research on the comparative effectiveness of different interventions. Therefore, this study addresses the following question: "How do crisis interventions affect mental health outcomes in emergencies?".

### Methods

This study aims to conduct a scoping review of the impact mechanisms of crisis intervention on the mental health of witnesses or participants in the context of emergencies. The study encompassed a wide array of studies, emphasizing the efficacy of several crisis intervention modalities, such as psychological first aid and trauma-focused therapy.

### Results

Most of the existing results were based on hospitals, schools and communities as research scenarios. The findings revealed substantial beneficial effects on mental health outcomes, such as decreased symptoms of posttraumatic stress disorder (PTSD), anxiety, and depression. Nevertheless, discrepancies in the efficacy were observed depending on the nature of the emergency, the model of intervention, and the demographic variables. The study highlights the intricate nature of executing crisis interventions during emergencies, taking into account aspects such as cultural sensitivity, resource availability, and the necessity for customized approaches.

### Conclusion

Crisis interventions are crucial in reducing the detrimental effects on mental health caused by emergencies. However, additional focused and enduring investigations are

**Data availability statement:** All relevant data are within the manuscript and its Supporting Information files.

**Funding:** This study is funded by the Sichuan Compulsory Education High-Quality Development Research Center (Project Number: YWYB-2023-03), the Sichuan Primary and Secondary School Teachers' Professional Development Research Center (Project Number: PDTR2022-34), and Center for Education Research at Sichuan Province (Project Number: TER2022-012). The funders had no role in study design, data collection and analysis, decision to publish, or preparation of the manuscript.

**Competing interests:** The authors have declared that no competing interests exist.

necessary to understand their efficacy in various emergency scenarios and demographics. Therefore, future research can further enrich psychological crisis intervention methods and deepen research on the impact of crisis intervention on mental health.

## 1. Introduction

The global impact of emergencies on mental well-being is an increasingly important area of concern. The World Health Organization (WHO) emphasizes that emergencies, such as conflicts, natural hazards, or public health crises such as pandemics, substantially impact the mental health and overall well-being of populations [1]. These events often lead to a wide range of psychological problems, from immediate stress and sorrow to long-term conditions such as post-traumatic stress disorder (PTSD), anxiety, and depression [2]. Charlson et al. [3] reported that 22% of people affected by conflict suffer from psychological disorders, highlighting the severe psychological burden of such events. Moreover, the World Health Organization (WHO) [4] reported that during the first year of the COVID-19 pandemic, there was a 25% increase in the occurrence of anxiety and depression globally. This highlights the significant mental health toll of such crises, highlighting the critical importance of addressing mental health during the pandemic.

During emergencies—such as natural disasters, disease outbreaks (epidemics), and other crises involving civilian populations—it is crucial to implement strategies that restore normalcy and safeguard mental health. This study specifically focuses on noncombat, civilian emergencies and excludes conflict-related settings such as war zones, genocides, or terrorist attacks. While armed conflicts such as those in Ukraine and Gaza can also result in public health emergencies due to mass displacement and healthcare system disruption, the present review focuses on interventions applicable to peaceful civilian contexts. These include epidemics (i.e., infectious disease-related emergencies), natural hazards, and community-based crises. To prevent ambiguity, the term "epidemic" is used throughout the study to refer exclusively to infectious public health events, excluding those arising from armed conflict. This scope allows for a more focused analysis of mental health interventions in nonconflict emergency settings.

Emergencies in humanitarian contexts significantly affect mental health and psychosocial well-being [5]. The COVID-19 pandemic has raised broad health concerns and worsened global mental health issues, revealing the profound psychological effects of such crises [6]. Jing et al. [7] highlighted how public health emergencies impact the well-being of individuals and communities. Their study emphasized the need for strong mental health support systems to meet growing demands after disasters [8]. These global trends point to an urgent need for effective crisis interventions that address both short- and long-term mental health challenges. As these emergencies become more frequent and severe, coordinated international efforts are essential to make mental health a key part of emergency response and recovery [9].

The recognition of crisis intervention in mental health during emergencies as a vital component of the global health response is growing. According to Muhamad et al. [10], crisis intervention is a short-term management approach designed to reduce the risk of lasting psychological harm for individuals affected by crises. Adopting this method is crucial for addressing current mental health problems and averting potential long-term repercussions. The World Health Organization highlights that individuals suffering from serious mental problems are especially susceptible to emergencies, thus requiring access to mental health services in addition to other essential necessities [1]. Crisis intervention provides essential mental health support during severe and unexpected emergencies, helping individuals cope with intense psychological distress [11]. Crisis intervention within emergency response frameworks focuses on acute mental health requirements and enhances the resilience and recovery of impacted populations [12].

The 'Emergency Response Law of the People's Republic of China' categorizes emergencies as sudden occurrences such as natural disasters, accidents, public health incidents, and social security incidents. These events can cause or potentially lead to serious social harm, requiring immediate response measures. In response to such emergencies, individuals often experience a range of nonspecific physiological and psychological reactions [13]. Most of these reactions are negative and maladaptive, leading to a divergence and differentiation in their emotions and psychology [14]. This may lead to psychological crises characterized by depression, anxiety, extreme stress, and fear. Unlike any other disaster, psychological crisis can inflict sustained and profound distress on individuals. Research indicates that active intervention, guidance, and treatment are crucial in assisting individuals in navigating these periods of mental distress [14]. The methods of crisis intervention vary depending on the type of crisis event, and there are different forms of mental health manifestations [15]. There is a scarcity of comprehensive studies reviewing the relationship and impact between crisis intervention and mental health, indicating a need for further exploration and clarification.

Contemporary scholarly works in crisis intervention during catastrophes have progressively emphasized the assessment of the efficacy of different intervention approaches and comprehending their influence on mental well-being results. These studies offer useful insights into the actual implementation and effectiveness of crisis interventions in various emergencies [16–18]. For example, Vernberg et al. [19] highlighted the importance of implementing multidisciplinary psychological crisis interventions during emergencies. This study emphasizes the importance of preserving the psychological well-being of the general population during times of crisis and investigates different approaches to accomplish this goal. This study indicates that customized interventions, taking into account the distinct requirements and circumstances of impacted groups, are essential for effectively treating mental health needs in emergencies. Devaskar et al. [20] examined the impact of crisis intervention on improving outcomes in patients with psychiatric disorders, specifically in emergency settings. Previous research has indicated that prompt and suitable crisis assistance might greatly enhance mental health results for those experiencing severe psychiatric distress. This study highlights the cost-effectiveness of crisis intervention strategies and their role in alleviating pressure on healthcare systems, especially during major events. Kerle [21] examined the effectiveness of crisis services in state Medicaid programs in diverting individuals from psychiatric hospitalization and minimizing the necessity for more extensive intervention in behavioral health crises [21]. This study emphasizes the cost-effectiveness of crisis intervention tactics and their role in reducing strain on healthcare systems, particularly during significant events. However, the literature also highlights deficiencies in research, both in terms of long-term results and the specialized requirements of specific populations. There is a demand for more extensive research that evaluates not only the immediate effects of crisis interventions but also their long-term efficacy in enhancing mental health outcomes.

The execution of crisis intervention during emergencies involves a variety of intricate issues and obstacles. These concerns not only affect the efficacy of interventions but also pose substantial obstacles in the administration and coordination of emergency mental health services. Ziegler [22] highlighted that a key obstacle in emergency management is effectively organizing and overseeing resources and services in times of disaster. This encompasses challenges related to information management, resource allocation, and integrating mental health care into broader emergency response efforts. The intricacy of overseeing these areas can greatly impact the prompt and efficient provision of crisis intervention

services. Team coordination poses a significant problem in the context of crisis management [23]. Issues such as information mishandling and poor resource distribution can hinder the efficiency of crisis response teams [24], particularly those providing mental health support. The absence of coordination can result in intervention delays, misallocation of resources, and ultimately, less-than-optimum mental health outcomes for individuals impacted by the emergency. Another significant challenge, as highlighted by the Substance Abuse and Mental Health Services Administration [25], is managing stress and promoting well-being among disaster response personnel. Responders, especially mental health professionals, frequently encounter elevated levels of stress, which can result in irritation, impaired task performance, and more severe problems such as escalated alcohol consumption or reliance on other coping strategies. This not only impacts their overall welfare but also hinders their capacity to deliver efficient crisis intervention. These challenges underscore the complexity of implementing crisis interventions in emergency settings. To address these problems, it is necessary to adopt a comprehensive strategy that involves enhancing coordination and communication, allocating sufficient resources, and prioritizing the mental health and well-being of emergency responders [26]. Overcoming these obstacles is vital for the efficient provision of crisis intervention services and for adequately addressing the mental health requirements of affected populations [27].

There is a notable lack of studies on crisis intervention during disasters, specifically in terms of applying theoretical knowledge and training to bring about tangible behavioral changes among rescuers in real-life situations. Research highlights the importance of examining how changes in crisis responders' attitudes and skills are related to their effectiveness in real-life emergency situations [28,29]. Moreover, there is a significant lack of comprehension of the subjective encounters of individuals undergoing mental health crisis care. The literature often overlooks the qualitative aspects that capture the real-life experiences of those affected by crises. Moreover, thorough research assessing the effectiveness of crisis response and safety planning, particularly in the realm of suicide prevention, is lacking. This highlights the urgent need for targeted studies in these areas. It is crucial to address these deficiencies to make progress in the field of crisis intervention and to guarantee that solutions are both efficient and adaptable to the demands of the individuals they intend to assist. This study analyzes and consolidates the current literature regarding the effects of crisis interventions on mental well-being in emergency environments. This entails the identification of efficacious methodologies, obstacles, and deficiencies in existing research, as well as comprehending the impact of various therapies and contextual variables on mental health outcomes.

This study presents various innovative contributions to the discipline. First, it offers a comprehensive analysis of the effects of crisis interventions on mental health during emergencies, a subject that has not been extensively examined in prior research. Second, it provides valuable perspectives on the relative efficacy of various crisis intervention approaches, addressing a significant deficiency in current studies. Finally, the study emphasizes the importance of contextual and demographic elements in influencing the results of crisis interventions for mental health outcomes and interventions within noncombat civilian emergencies.

## 2. Materials and methods

### 2.1 Scope overview

This study focuses solely on crisis counseling, as it pertains to civilian situations when public health emergencies or natural disasters occur, ignoring the counseling that relates to military and war zones. This context allows us to focus on the strategies used to address mental health in nonwar situations—places that are not defined by armed conflict, genocide, or terrorism.

Scoping reviews are distinct from conventional or systematic literature reviews in that they are better suited for broader research questions, aiming to describe an overarching view of a given topic [30]. Additionally, scoping reviews follow a structured process and rigorous design under the guidance of a research protocol (Table 1), reducing the reliance on a researcher's subjective knowledge and experience, as is often the case in traditional literature reviews [31]. This method allows for the compilation of more comprehensive literature. It helps researchers understand the overall landscape and limitations of existing research, thereby guiding the direction for future studies.

**Table 1. Research protocol.**

| Item | Details |
|------|---------|
| Main focus | The study focuses on:<br>• Methods and applications of crisis intervention for mental health during emergencies.<br>• Indicators and assessment tools for evaluating mental health in such contexts.<br>• The relationship and impact mechanisms linking crisis intervention to mental health outcomes. |
| Approach | Based on PRISMA-ScR checklist |
| Methods | - Literature Search: Web of Science, and MEDLINE<br>- Inclusion/exclusion criteria<br>- Data extraction<br>- Quality assessment<br>- Data synthesis: Narrative approach |
| Publication period | From database inception to September 2023 |
| Keywords | Crisis intervention, psychosocial interventions, psychosocial consequences, mental health, mental well-being, emergency, acute crisis, PTSD, anxiety, depression, trauma, traumatic even, stress |
| Inclusion criteria | (a) The research must utilize original data and not be review articles;<br>(b) The content of the research should be related to the impact of crisis intervention on mental health of civilians, rather than focusing solely on the use of a specific crisis intervention method;<br>(c) The research should be quantitative, employing scales to assess mental health outcomes both before and after the intervention. |
| Exclusion criteria | (a) Research for which the full text is not available; (b) Studies that do not elucidate the impact of crisis intervention on mental health; (c) Research that does not use a scale to measure mental health outcomes; (d) Studies that do not report the period during which the research was conducted; (e) Research not based on sudden onset events as the background; and (f) Non-English/Chinese articles, reviews, studies not focusing on crisis intervention. |
| Regional focus | Global, with emphasis on studies from diverse geographical regions. |

## 2.2 Research focus

The primary concerns of this study are (1) the main methods and applications of crisis intervention for mental health during emergencies; (2) specific indicators and assessment tools used for evaluating mental health during emergencies; and (3) the relationship between crisis intervention and mental health in the context of emergencies, along with its underlying impact mechanisms.

## 2.3 Literature search methods

This study used the Web of Science (WOS) core collection and the MEDLINE database to conduct quantitative research searches in English. The search period covered the time from each database's inception to September 2023, when the research team completed the searches. The search string for English literature was "Article Title (Crisis Intervention) AND Article Title (Mental Health or PTSD or Psychological crisis or Depression or Anxiety or Trauma or Stress or Disorders or Emotion)".

## 2.4 Literature screening criteria

Several criteria have been followed for initial screening inclusion: (a) the research must utilize original data and not review articles; (b) the content of the research should be related to the impact of crisis intervention on mental health rather than focusing solely on the use of a specific crisis intervention method; and (c) the research should be quantitative, employing scales to assess mental health outcomes both before and after the intervention. Similarly, several criteria have been followed for second screening exclusion: (a) research without full-text availability; (b) studies that do not elucidate the impact of crisis intervention on mental health; (c) research that does not use a scale to measure mental health outcomes; (d) studies that do not report the period during which the research was conducted; and (e) research that does not focus on sudden onset events as the background.

## 2.5 Literature screening process

The literature screening process was conducted in three stages: retrieval, primary screening, and secondary screening. Initially, a database search yielded 241 documents. During the primary screening stage, three researchers trained in systematic review methods conducted an independent preliminary review. They read the titles and abstracts to eliminate irrelevant studies and then cross-compared and collectively discussed their findings to reach a consensus. This stage resulted in the exclusion of 29 duplicate documents, leaving 212 articles for secondary screening. Two researchers independently reviewed the full texts during the secondary screening, adhering strictly to the exclusion criteria. In cases of uncertainty, discussions were held among the researchers to reach a consensus. After the final secondary screening, 54 studies were excluded, and 158 studies remained. Further exclusions were made for reasons such as the unavailability of the full text or unspecified research periods, leading to an additional 76 documents being excluded. Ultimately, 82 articles were selected for information extraction and summary analysis. The detailed literature screening process is illustrated in Fig 1.

## 2.6 Information extraction and analysis

This study meticulously adhered to the methodological framework of a scoping review, referencing the steps outlined by Arksey and O'Malley [32] and incorporating improvement suggestions from Levac and others. The compilation and summarization of information were conducted in line with the reporting guidelines for scoping reviews. The extracted information encompassed various aspects, including types of crisis events, research settings, timelines of research development,

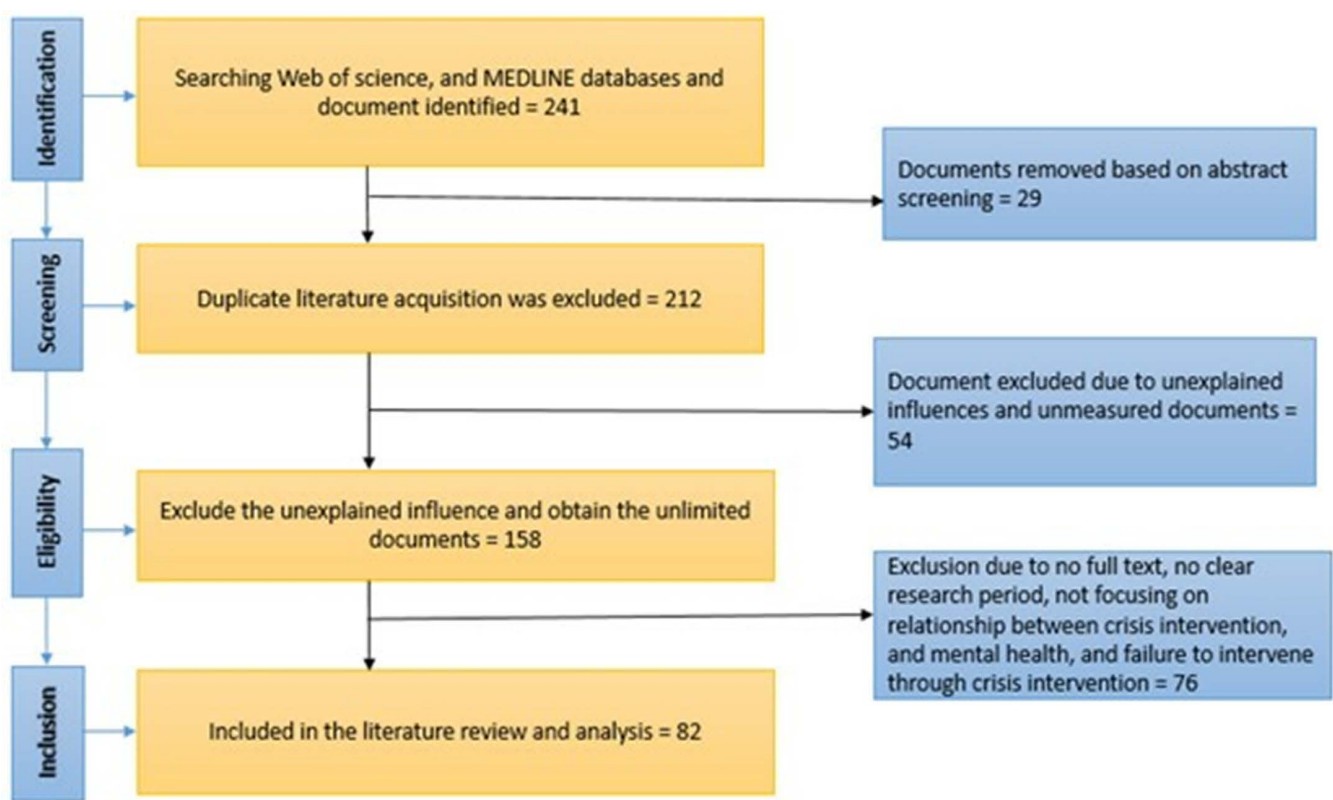

**Fig 1. Search strategy and results flow chart.**

research designs, methods of crisis intervention and their effectiveness, specific mental health indicators and assessment tools, the relationship between crisis intervention and mental health, and the mechanisms and pathways of impact.

## 2.7 Quality assessment,

To ensure the reliability and accuracy of scoping reviews, quality assessment is essential. Quality assessment was incorporated into this scoping review to contextualize and interpret the findings. Our analysis of the methodological rigor and potential biases of the included studies allowed us to gain a deeper understanding of the outcomes. Studies were evaluated on the basis of their design, implementation, and documentation. The quality evaluation results enabled us to identify research that employs rigorous techniques, thus enhancing the credibility of our synthesis findings. However, research with methodological deficiencies was carefully interpreted. A scoping study conducted by Springer in 2023 revealed that quality evaluation methods are useful for determining the reliability of observational studies, particularly in fields where research is in its early stages [33]. Rather than excluding research, the quality assessment outcomes offered a discerning perspective to better understand the topics reviewed, ensuring a thorough appreciation of the findings.

## 2.8 Data analysis and synthesis

This study employed a narrative synthesis approach, which is well suited for scoping reviews because of its flexibility in incorporating diverse study designs and methodologies. This approach facilitated the incorporation of numerical and descriptive information, providing a comprehensive perspective of the research terrain. The results of this scoping review were presented methodically and easily. Tables were utilized to briefly present the fundamental attributes of the included studies, such as the study's structure, sample size, and primary results. Charts were used to visually depict data patterns, such as the frequency of various mental health outcomes of civilians after crisis interventions. In addition, thematic summaries were also provided to highlight key findings and important themes from the data.

# 3. Results and analysis,

## 3.1 Including the basic characteristics of the literature

This scoping review covers a range of civilian nonmilitary emergency settings. These include communities, hospitals, schools, and public health institutions. It explicitly excludes crisis interventions within conflict zones, in genocide-affected areas, or in direct response to terrorist attacks, which considerably narrows the focus of the work to mental health outcomes in nonconflict, civilian emergencies.

Table 2 displays the basic characteristics of the 120 documents included in this study. The research settings covered a range of environments, such as communities, schools, hospitals, police stations, and fire stations. The populations in the selected studies vary widely, including different demographics, such as those of students, children, patients with mental illnesses and their families, mental health workers, police officers, accident victims, firefighters, and COVID-19 patients. These diverse populations provide a comprehensive understanding of the effectiveness of crisis interventions across various groups affected by emergencies. The sample sizes in these studies varied, with the smallest being 22 participants, the median being 328, and the largest involving 2120 participants. The research designs employed in these studies included cross-sectional comparative surveys, longitudinal follow-up surveys, and mixed-method surveys.

## 3.2 Crisis intervention methods and usage

Crisis intervention is recognized for its effectiveness in preventing diseases, alleviating symptoms, reducing comorbidities, and preventing their prolongation [18]. It is characterized as short-term, timely, and effective and is widely used to treat psychological crises during emergencies. However, owing to variations in the types of emergencies and corresponding mental health issues, research has employed multiple crisis intervention methods.

**Table 2. Summary of basic characteristics of the included studies.**

| Theme | Feature | Count | Percentage |
|---|---|---|---|
| Incident type | Terrorist attacks | 8 | 6 |
| | COVID-19 | 21 | 18 |
| | Traffic accident | 23 | 19 |
| | Coal mine accident | 9 | 8 |
| | Hurricane | 13 | 11 |
| | Community disaster | 6 | 5 |
| | Public health incident | 18 | 15 |
| | Social security incident | 22 | 18 |
| Research scenario | Community | 27 | 23 |
| | School | 25 | 21 |
| | Hospital | 31 | 26 |
| | Police station | 17 | 14 |
| | Fire station | 13 | 11 |
| | Others | 7 | 5 |
| Intervention time | Intervention 0-Jan | 20 | 17 |
| | Intervention January-March | 21 | 18 |
| | Intervention March-June | 27 | 23 |
| | Intervention June-December | 19 | 16 |
| | Intervention 1–2 years | 12 | 10 |
| | Intervention 2–3 years | 10 | 8 |
| | Intervention for more than 3 years | 11 | 9 |
| Sample size | Minimum value | 2 | 2 |
| | Median | 32 | 8 |
| | Maximum value | 21 | 20 |
| Research design | Cross-sectional comparative survey | 77 | 64 |
| | Longitudinal follow-up survey | 13 | 11 |
| | Mixed studies | 30 | 4 |

Among the literature reviewed, 47 articles implemented the crisis intervention team (CIT) method, also known as the "memphis model" [34]. This model, first introduced by the Memphis police department in 1988, primarily involves providing mental health training to various adult groups. The training includes lectures, experiential learning, and visits to treatment facilities and uses various scales to assess participants' views on mental health services throughout the training process. Notably, one article focused on the crisis intervention team-young (CIT-Y), an adaptation of the CIT model that tailors mental health services for teenagers [35]. This approach enables police officers to modify their interactions and attitudes toward adolescents with mental health issues.

Several articles adopted the Neuman health system model [36–38]. The central tenet of Neuman's theory is to view individuals as holistic, open, multidimensional systems that continuously interact with environmental stressors. The theory posits that through their defense systems, individuals maintain the balance and integrity of the entire system. During interventions, separate experiments are conducted with observation and control groups. Neuman health care system theory is applied to intervene in the observation group across five dimensions: physiological, psychological, sociocultural, developmental, and spiritual.

Several articles utilized the Satir Model, also known as the Satir Communication Model, developed by Virginia Satir, a pioneer in family therapy in the United States [39,40]. This model employs a comprehensive theoretical system to

address individual psychological issues. Techniques such as iceberg theory, communication stances, and family restructuring are used to intervene with patients for varying periods after a crisis. Tools such as the "Four Coping Postures" aid in understanding communication patterns with others, whereas the "Family Relationship Diagram" and "Wheel of Influence" help individuals comprehend their roles within families or social circles. Some studies have explored the use of informal response mechanisms combined with short-term assessment interviews [41,42]. These informal mechanisms often involve support from caregivers, family, friends, or peer networks. The intervention, typically consisting of one or more individual or group sessions, is conducted within hours or days of a traumatic event, utilizing short-term assessment interview tools.

Several articles have examined three-stage online psychological crisis intervention programs [11,43]. This intervention spans three distinct periods: the day before entering an isolation ward (time 1), the first day after leaving the isolation ward (time 2), and the end of the intervention (time 3). The program measures and compares mental health issues across these different timeframes. Another five articles discussed the "online plus offline" method [44–48]. Following a crisis, medical staff conducted initial online telephone interviews supplemented by offline face-to-face treatment. Additionally, several articles focused on an "emergency services" approach [49–53]. In this method, each new patient receives an initial assessment from a paramedic team member to determine the required level of treatment. The immediate services provided include clinical assessment, treatment, patient education, and medication management for crisis intervention via telemedicine.

Several articles explored diverse approaches to psychological crisis intervention, each with its own unique methodology and focus. Telephone therapy was utilized in some studies [48,54,55], where interventions were conducted exclusively through phone calls. This approach, while flexible, lacks a standardized plan and specialized training, making the effectiveness of the interventions heavily reliant on the therapists' professional experience. A four-step psychological crisis intervention, detailed in three articles, is typically executed postemergence and includes stages of self-introduction, the expression of feelings, the provision of information to mitigate negative emotions, and the establishment of effective support through targeted training to bolster coping mechanisms.

The dual-factor model of mental health (DFM), explored in another set of three articles [56–58], highlights the importance of positive psychological states by using subjective well-being as a positive indicator and psychopathology as a negative indicator. This model categorizes patients into four distinct groups on the basis of their mental health status and adapts scales to suit various crises. Critical event stress management (CISM), combined with training courses detailed in three articles, is implemented several months after an emergency [12,17,59]. This method assesses stress levels through questionnaires and customizes course training to address identified needs. Additionally, a clinical care process involving initial triage by a mental health crisis consultant in the emergency department is described in three articles. This process assesses the immediate need for medical evaluation, considering factors such as medical illness, alcohol/substance use, and consciousness levels.

Two articles introduce the Weiss crisis intervention model, which focuses on imparting practical crisis intervention knowledge and skills through simulated scenarios [60,61]. This model encourages learners to take on ally roles and implement crisis intervention strategies in group settings for hands-on experience. Other methods documented include postincident health monitoring, mental health crisis intervention services, online interventions, decision-making groups paired with civilians' mental health interventions and intelligent disease monitoring systems, short-term assessment interviews for posttraumatic stress disorder, mental health crisis response teams, crisis management systems, and child protective services (CPSs), among others. These varied approaches, each with its specific application and theoretical underpinning, are detailed in Table 3, highlighting the wide range of strategies employed in addressing psychological crises.

### 3.3 Specific indicators and assessment tools for mental health

Following an emergency, individuals often experience intense psychological tension, leading to psychological imbalances that can severely impact their mental health. Literature review findings show that mental health problems can

**Table 3. Crisis intervention methods and usage.**

| Intervention | Description | Sources |
|---|---|---|
| Crisis intervention team (CIT) | Adult-focused training programs. Mental health training is provided primarily through intensive lectures, experiential training, and visits to treatment facilities. | [16,28,29,35,62–68] |
| Neuman health system model | During the intervention process, the observation and control groups were used for separate experiments. Neuman's health care system theory was used to intervene in the observation group in five aspects: physiology, psychology, social culture, development direction, and spirit. | [38] |
| Satir therapy model | After the crisis event occurs, the patient is intervened for varying periods through interview sampling and scales. | [38] |
| Informal response mechanism + short-term evaluation interview | An intervention consisting of 1 or more individual or group sessions targeted within hours or days of a traumatic event through the use of the Short Assessment Interview Tool | [18,42] |
| Three-stage online psychological crisis intervention program | The intervention was carried out at three different periods and its results were assessed against measurement data. | [11,18] |
| Online and offline | After the crisis occurs, online phone calls and offline face-to-face interviews are used, and different scales are used to intervene offline based on the content of the online phone calls. | [44–48] |
| Emergency services | Each new patient undergoes an initial assessment by a member of the paramedical team to determine the level and type of treatment required before receiving a psychiatric assessment by a senior psychiatrist. | [49,51–53] |
| Phone therapy | Intervention is carried out through a single telephone call. There is no unified plan and no special training. The professional experience of each therapist will affect the effect of the intervention. | [48,54,55] |
| Four steps of psychological crisis intervention | The intervention is divided into four steps, including self-introduction, expressing feelings, information to prevent negative emotions, and establishing effective support through appropriate training to improve coping abilities. | [69] |
| Dual-Factor Model of Mental Health (DFM) | Initiate clinical care intervention with triage and have a mental health crisis counselor in the emergency department to assess the need for medical illness, alcohol/substance use, and consciousness to determine the need for medical evaluation | [56–58] |
| Critical Incident Stress Management (CISM) + Training Course | Conducted several months after the emergency, the stress module is assessed through a questionnaire, and course training is conducted based on the questionnaire results. | [12,17,59] |
| Clinical care model | Initiate clinical care intervention with triage and have a mental health crisis counselor in the emergency department to assess the need for medical illness, alcohol/substance use, and consciousness to determine the need for medical evaluation | [70–72] |
| Weiss crisis intervention model | Offer yourself as an ally to the couples in the group and facilitate them expressing support for each other. Further planning includes creating an atmosphere of acceptance of the feelings evoked by this event, with the expectation that these healthy individuals will be able to use the group and their spouses to cope with this stressful moment effectively | [60,61] |
| Other | Including postincident health monitoring, MH crisis intervention services (mental health care (MH)), online intervention, decision-making group + establishment of public mental health intervention + intelligent disease monitoring system, posttraumatic stress disorder short-term assessment interview, mental health crisis response Intervention models such as groups, crisis management systems, and Child Protective Services (CPS) | [5,11,73–77] |

manifest in various forms, primarily as symptoms of anxiety, depression, psychological trauma, and stress. Among these, posttraumatic stress disorder (PTSD) was the most frequently addressed issue (n=28), followed by general psychological problems (n=18) and depression (n=13). The studies utilized various scales or international diagnostic standards to assess mental health changes accurately. These tools were employed to measure and compare mental health status before and after the interventions. The specific findings from these assessments are presented in Table 4.

**Table 4. Mental health-related indicators and assessment tools.**

| Mental health indicator | Metric count | Assessment tools | Description | Sources |
|---|---|---|---|---|
| PTSD | 28 | SF-36, IES-R, BSI, SCL-90, TDS, PTSD-10, HoNOS, DSM-5, PCL-5 | PTSD (Post-Traumatic Stress Disorder) is assessed using various tools such as the SF-36 Health Survey, Impact of Event Scale-Revised (IES-R), Brief Symptom Inventory (BSI), Symptom Checklist-90 (SCL-90), The Social Distance Scale (TDS), The Posttraumatic Symptom Scale (PTSD-10), Health of the Nation Outcome Scales (HoNOS), Diagnostic and Statistical Manual of Mental Disorders – Fifth Edition (DSM-5), and Posttraumatic Stress Disorder Checklist – Fifth Edition (PCL-5). | [2,23,78–87] |
| Psychological problems | 18 | SCL-25, SGCMHS, DTS, SF-36 | General psychological problems, including anxiety and depression, are assessed using tools such as the Symptom Checklist-25 (SCL-25), Stigma in Global Context - Mental Health Survey (SGCMHS), Distress Tolerance Scale (DTS), and the SF-36 Health Survey. | [18,53,88–93] |
| Depression | 13 | CDI, BDI, HDRS, MADRS, QIDS-SR | Depression is assessed using tools such as the Children's Depression Inventory (CDI), Beck Depression Inventory (BDI), Hamilton Depression Rating Scale (HDRS), Montgomery-Asberg Depression Rating Scale (MADRS), and the Quick Inventory of Depressive Symptomatology – Self-Report (QIDS-SR). | [84,85,94–102] |
| Psychological trauma | 12 | MADRS, QIDS-SR | Psychological trauma is evaluated using tools such as the Montgomery-Asberg Depression Rating Scale (MADRS) and the 16-Item Quick Inventory of Depressive Symptomatology – Self-Report (QIDS-SR). | [93,103–107] |
| Post-Covid trauma | 12 | BSTE, SRQ-20 | Post-COVID trauma is assessed using the Brief Scale of Triage in Emergencies (BSTE) and the Self-Reporting Questionnaire 20 (SRQ-20). | [43,45,82,90,96,108,109] |
| Insomnia | 10 | CD-RISC, PSS, PSQI, ISI | Insomnia is evaluated using tools such as the Connor-Davidson Resilience Scale (CD-RISC), Chinese 14-item Perceived Stress Scale (PSS), Pittsburgh Sleep Quality Index (PSQI), and Insomnia Severity Index (ISI). | [11,14,53,89,100,101,110–113] |
| Anxiety | 8 | GAD-7, PHQ-D, SAS, QOL | Anxiety is assessed using tools such as the Generalized Anxiety Disorder 7-item (GAD-7), Patient Health Questionnaire (PHQ-D), Self-Rating Anxiety Scale (SAS), and Quality of Life (QOL). | [2,11,82,96,100,101,114–117] |
| Adjustment disorder | 8 | AAQ-II, SSS, SCL-90 | Adjustment disorder is evaluated using tools such as the Acceptance and Action Questionnaire – 2nd Edition (AAQ-II), Somatic Self-Rating Scale (SSS), and Symptom Checklist-90 (SCL-90). | [46,89,118] |
| Emotional instability | 8 | EIS, DASS-21, AAQ-II, ESS | Emotional instability is assessed using tools such as the Emotional Intelligence Scale (EIS), Depression Anxiety Stress Scales (DASS-21), Acceptance and Action Questionnaire – 2nd Edition (AAQ-II), and Eysenck's Emotional Stability Scale (ESS). | [45,46,119] |
| Stress | 3 | PSS-10, PHQ-D, WHOQOL-BREF, DTS | Stress is evaluated using tools such as the Perceived Stress Scale (PSS-10), Patient Health Questionnaire (PHQ-D), World Health Organization Quality of Life Scale – Brief Form (WHOQOL-BREF), and Distress Tolerance Scale (DTS). | [2,53,100,115,120] |

### 3.4 Relationships between crisis interventions and mental health

Keywords provide a high-level summary, capturing the core ideas and themes of the literature included in this study. This article aims to cluster these keywords to illustrate the relationship between crisis intervention and mental health research. As shown in Fig 2, VOSviewer software was used for a co-occurrence analysis of the keywords. The data from the selected literature were imported into VOSviewer, and a threshold of 18 was set for conducting the keyword cluster analysis. This analysis generated four distinct keyword clusters. The most frequently identified keywords were mental health, crisis intervention, COVID-19, anxiety, depression, PTSD, trauma, adolescence, and stress.

First, research on specific indicators of mental health in emergencies extensively covers terms such as mental health, anxiety, depression, PTSD, and trauma [98,101,117]. This body of work primarily delves into the mental health indicators

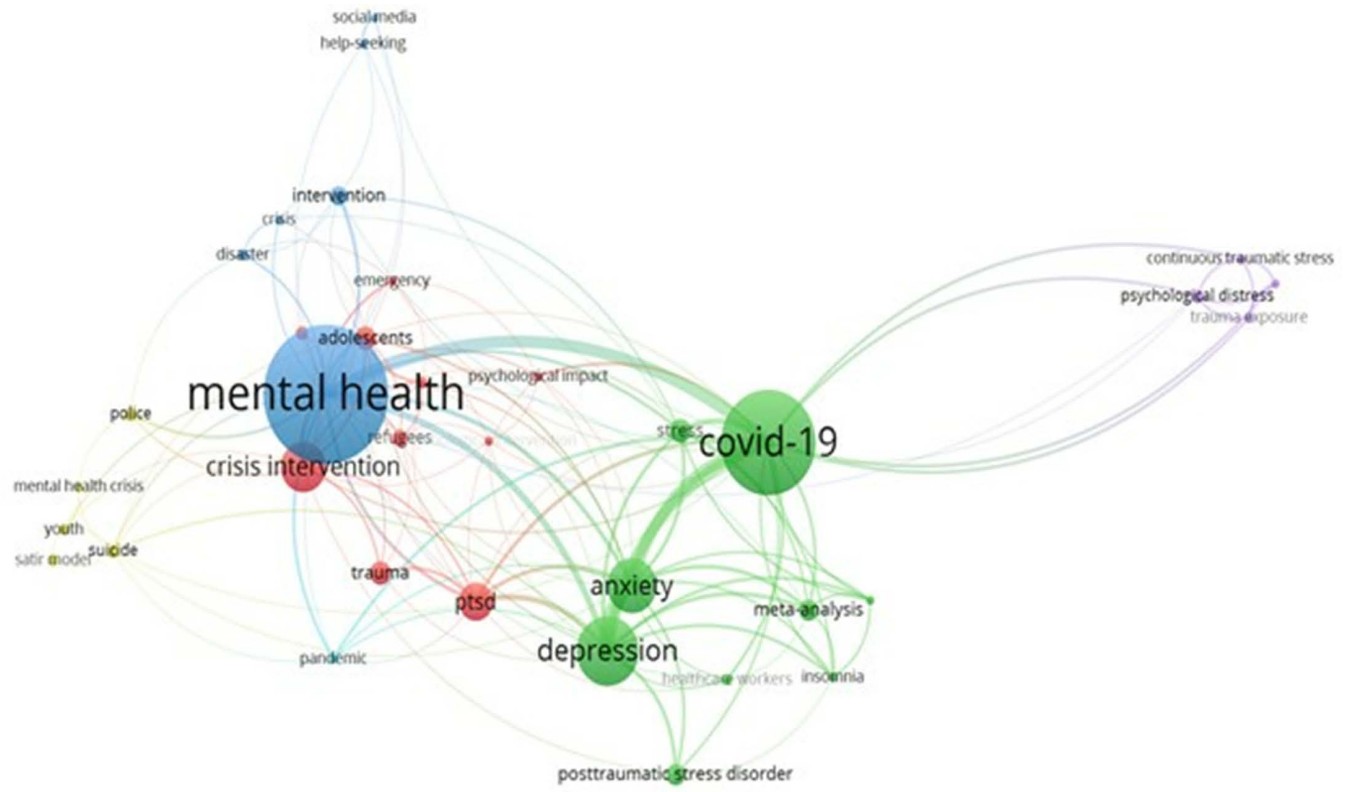

**Fig 2. Research literature keyword co-occurrence network map.**

prevalent during emergencies, where anxiety and depression emerge as common psychological responses to crisis events. The term 'COVID-19' highlights the significant impact of the global public health crisis.

Second, keywords related to crisis interventions, such as interventions, emergencies, and disasters, suggest the need for civilians experiencing mental health issues due to crisis events to undergo assessments and receive appropriate interventions [70,121,122]. These terms highlight the importance of timely and effective crisis intervention methods.

Third, in the context of specific population groups and intervention settings, key terms such as *adolescents, healthcare workers,* and *refugees* highlight the emphasis on vulnerable populations. The inclusion of adolescents and healthcare workers highlights the importance of developing tailored interventions, as these groups are especially susceptible to psychological distress during crises [71,123].

Finally, the interplay among these keywords highlights strong interconnections between the themes, illustrating a holistic crisis intervention process aimed at improving mental health in the aftermath of an emergency. The visualization indicates that mental health and crisis intervention are central themes closely linked with COVID-19, anxiety, depression, PTSD, and trauma.

### 3.5 The impact mechanism and path of crisis intervention on mental health

Analysis and statistics from the included literature revealed that crisis intervention generally positively influences mental health during emergencies, operating through both direct and indirect pathways.

**3.5.1 Direct impact of crisis intervention on mental health.** Numerous studies have shown that crisis intervention has a direct effect on mental health. Various crisis intervention methods and approaches have been shown to effectively

mitigate the psychological crisis of civilians. For example, Yang [124] examined 22 quarantine personnel in a designated quarantine hospital in Mianyang from February 4--29, 2020. Employing an "online + offline" intervention model for a 2-week psychological intervention, the HAMD-24 and HAMA scores of the quarantined personnel were significantly lower on the first and second weekends than before the intervention began. These findings indicate that the "online + offline" psychological intervention model can effectively reduce anxiety and depression among quarantined individuals.

**3.5.2 Indirect impact of crisis intervention on mental health.** In emergencies, while appropriate crisis intervention methods can directly benefit mental health, some scholars have also highlighted that crisis intervention can indirectly improve mental health by affecting other variables. Guanhua [125] noted that in special emergency scenarios, appropriate psychological crisis intervention measures can aid individuals in properly confronting difficulties and setbacks. This strategy promotes the development of psychological resilience in the face of emergencies, thus diminishing or neutralizing the adverse effects of such challenges.

## 4. Discussion

### 4.1 Effectiveness of crisis intervention

The outcomes highlight the efficiency of different crisis intervention techniques, such as psychological first aid and trauma-focused methods, which are particularly relevant in emergencies with no direct conflict involving civilians. The analysis focuses on public health interventions within civilian contexts, juxtaposing them with a selection of nonpublic health-related actions. The intent is to demonstrate that when injury types are similar, public health emergencies warrant a unique and unprecedented set of nonviolent interventions to mitigate community harm. It is within this framework that the analysis does not engage military public health interventions.

The findings suggest that various crisis intervention modalities, such as psychological first aid and trauma-focused treatment, are associated with positive mental health outcomes, including reductions in symptoms of PTSD, anxiety, and depression. These associations are consistent with previous studies, such as the work of Mediavilla et al. [126], which reported decreases in anxiety and depression among healthcare professionals during the COVID-19 pandemic.

The literature offers a detailed perspective on the complex nature of crisis response in the context of specific civilian emergencies across different approaches. Marcus and Stergiopoulos [127] performed a swift review of results between police, coresponder, and nonpolice approaches in mental health crisis response. Their research indicates that crisis intervention teams (CITs) and coreponder models, which combine law enforcement personnel with mental health physicians, provide potential benefits. However, the data concerning their effects on crisis outcomes are variable and sometimes ambiguous. This emphasizes the need for a more detailed understanding of the operations of different crisis intervention models in diverse emergencies.

Wesemann et al. [17] noted that emergency responders' mental health outcomes after terrorist attacks varied depending on whether they received crisis intervention, emphasizing the importance of tailoring interventions to specific groups. Research findings have indicated that crisis intervention may not have consistent positive effects on all people, as some subgroups showed worse quality of life and increased depression symptoms after the session. This discovery emphasizes the need to customize crisis responses to address the distinct requirements of various groups, taking into account variables such as gender and employment.

Manchanda et al. [128] studied how friendship interventions might impact the mental health of teenagers, highlighting the need for social support during crises. Their comprehensive evaluation indicates that therapies including a buddy or genuine social group may have beneficial short-term impacts on teenagers' mental health, but the long-term impact is uncertain. This highlights the importance of using social networks and connections to create successful crisis solutions.

Austin et al. [129] enhanced the conversation by creating an algorithm to improve the efficacy of interventions for parents displaying indications and symptoms of mental health issues [4]. Their research demonstrated the potential of

data-driven methods to tailor treatments to individual needs, suggesting a way to enhance crisis intervention effectiveness through personalized care.

Various elements, such as the intervention model, the unique requirements of the target population, and emergencies, impact the success of crisis intervention. Some models are promising; however, the data highlight the importance of customizing treatments on the basis of individual and group characteristics to improve effectiveness. Future studies should focus on developing creative and individualized methods for crisis intervention to enhance mental health results during emergencies.

## 4.2 Factors influencing the efficacy of crisis interventions

Various factors impact the effectiveness of crisis interventions, including the type of emergency, the intervention model, and demographic characteristics. The study's results align with existing research, indicating that these characteristics influence variations in the effectiveness of crisis interventions. Evans et al. [130] highlighted the importance of adapting crisis interventions to meet the needs of children and young people. Similarly, the adaptability of crisis intervention teams (CITs) across different settings was emphasized by Kane et al. [131]. The nature of the problem, such as a natural catastrophe, a public health issue such as the COVID-19 epidemic, or a terrorist attack, might impact the efficacy of crisis solutions. The psychological effects of various events vary, requiring distinct crisis response strategies. The COVID-19 pandemic has emphasized the need for psychological crisis interventions that address not only the fear of infection but also the social isolation and economic uncertainty linked to the epidemic.

The effectiveness of the crisis intervention paradigm may significantly affect its effectiveness. Conventional approaches such as crisis intervention teams (CITs) have been evaluated for efficacy in different environments. An evaluation of the effectiveness of a crisis intervention team (CIT) at the Esplugues Mental Health Center in Barcelona investigated the potential benefits of such teams in certain situations. The success of crisis intervention team (CIT) models and other treatments, such as coresponder models, varies and is frequently influenced by a particular implementation and setting.

Demographic factors, such as age, gender, and cultural background, also play crucial roles in determining the effectiveness of crisis interventions. Differences in gender and profession have been observed in the mental health outcomes of emergency responders following a terrorist incident, indicating that crisis interventions should be tailored to address these differences effectively. Moreover, treatments aimed at certain demographic groups, such as teenagers, could need components that address the distinct issues these groups encounter, such as social isolation or academic expectations [4].

The effects of crisis interventions may be considerably influenced by how they are implemented and the environment in which they are provided. Factors such as resource availability, staff training, and cultural sensitivity may affect the success of interventions. Furthermore, the success of treatments relies heavily on their acceptability and accessibility to the target group.

The factors affecting the effectiveness of crisis interventions highlight the need for a detailed and customized strategy for crisis management. In addition to the nature of the emergency, the demographic traits of the impacted population and the intervention setting may aid in creating more efficient crisis solutions. Future research should fill these gaps by conducting longitudinal studies and exploring the processes driving the effectiveness of crisis interventions.

## 4.3 Research gaps

Most of these studies have conducted comprehensive quantitative analyses via rigorous measurement and evaluation tools, resulting in significant research findings. However, the existing research has several limitations, indicating potential directions for future studies.

**4.3.1 Expansion of research on psychological crisis intervention methods.** The literature review indicates that most research on crisis intervention spans up to one year. However, individuals affected by emergencies may experience not only acute stress disorders shortly after the event but also prolonged issues such as delayed posttraumatic stress

disorder, which can lead to chronic psychological trauma, personality changes, and significant adverse effects on physical and mental health, as well as on work and life. Thus, establishing a long-term psychological assessment and crisis intervention mechanism is essential and of significant research interest. Additionally, while most studies assess the effects of interventions through pre- and postintervention comparisons, there is a practical need to broaden the level and scope of psychological crisis intervention organizations. This involves combining efforts from various entities to create a comprehensive social psychological counseling mechanism. From an academic perspective, conducting multidimensional joint intervention research could further enrich the research landscape.

**4.3.2 Deepening process research on the impact of crisis intervention on mental health.** The efficacy of crisis intervention as a primary method for mitigating psychological crises is well established. The current findings are predominantly based on comparisons of pre- and postintervention assessments for witnesses and participants and outcomes from experimental and control groups. Most research to date has focused on answering the outcome question of "whether it is effective." However, there is a notable gap in the detailed examination of the mechanisms through which crisis intervention influences mental health, the evolution and variations in the timing and effects of actions, and the presence of consistent patterns throughout the intervention process. Addressing "how it is effective" represents a critical area for further in-depth investigation.

## 4.4 Policy implications

On the basis of the analysis of the effectiveness of crisis interventions and the factors influencing their success, there are clearly significant practical implications for mental health crisis management. These implications are crucial for policymakers, mental health experts, and emergency response teams seeking to improve the mental health results of those impacted by disasters.

**4.4.1 Tailored crisis intervention strategies.** The varying success of crisis interventions in various circumstances and demographics highlights the need for customized tactics that consider the unique aspects of the crisis, the cultural context, and the demographic traits of the afflicted community. Practitioners should possess various intervention tools and methods that may be customized to suit the specific requirements of each case. Interventions for a natural catastrophe may involve community-based assistance and the restoration of social networks. In contrast, interventions for a public health crisis such as COVID-19 may need to concentrate on treating isolation and economic stress.

**4.4.2 Training and Resources.** The results emphasize the need to offer sufficient training and resources to professionals engaged in crisis intervention. This involves receiving instruction in culturally sensitive methods, comprehending the distinct requirements of various groups, and successfully using a range of intervention strategies. To ensure effective intervention delivery, it is essential to provide crisis intervention teams with the required resources, such as mental health support tools and referral networks.

**4.4.3 Interdisciplinary Collaboration.** An efficient crisis response requires a cooperative strategy that includes many sectors, such as healthcare, social services, education, and law enforcement. Collaborating across disciplines may improve the thoroughness and efficiency of crisis solutions by merging knowledge from many professions. Coresponder models, which combine law enforcement personnel with mental health therapists, have shown potential in some situations and emphasize the advantages of collaborative strategies.

**4.4.4 Community engagement and support.** Involving communities in creating and carrying out crisis responses may improve their significance and efficiency. Community engagement is crucial for ensuring that treatments are based on the local context and culture, which is essential for their acceptability and effectiveness. Additionally, establishing community resilience and support networks may provide a lasting basis for mental health assistance after a crisis.

**4.4.5 Continuous evaluation and research.** Ongoing assessment and research are necessary to determine the efficacy of crisis interventions and identify optimal practices. This involves conducting high-quality research to investigate the long-term effects of interventions and the processes by which they influence mental health outcomes. Continued

research can inform the development of evidence-based strategies and policies that adapt to the changing dynamics of crises and the needs of affected communities.

The practical consequences of the debate on crisis intervention effectiveness and influencing factors emphasize the challenges of meeting mental health needs in the context of specific civilian emergencies. Practitioners may improve the efficiency of crisis interventions by using customized, innovative, cooperative, and community-oriented strategies. Continuous assessment and study are crucial for improving these methods and guaranteeing that they cater to the varied requirements of persons and communities impacted by disasters.

## 5. Conclusion

The relevant journal articles were analyzed via the PRISMA-ScR checklist for scoping reviews and a narrative review. Focusing on specific types of civilian emergencies, we highlight intervention strategies that can be tailored to nonconflict settings. This review covered 82 articles that focused on the impact of crisis interventions on mental health during emergencies. There is a noticeable trend of increasing scholarly interest in the effects of crisis intervention on individuals' mental health.

Our study revealed that crisis interventions play a crucial role in enhancing mental health outcomes during catastrophes by notably reducing symptoms of posttraumatic stress disorder (PTSD), anxiety, and depression across different populations and emergency scenarios. The efficiency of these interventions depends on several aspects, such as the kind of emergency, the particular intervention model used, and the demographic features of the afflicted community. The results highlight the need for adaptable, culturally sensitive, customized crisis interventions to address the specific needs of individuals and communities facing disasters.

This study emphasizes the need for emergency response frameworks that integrate adaptable and evidence-based crisis intervention strategies that can be tailored to an event's unique circumstances and the impacted community's demographics. This research highlights the importance of multidisciplinary teamwork and community engagement in creating and executing crisis interventions to improve their efficacy and approval.

This study has several limitations. First, the exclusion of conflict settings, while intended to maintain a specific focus on natural hazards and public health emergencies, limits the generalizability of the findings. Second, the reliance on literature up to September 2023 may exclude recent developments. Third, broadening keywords resulted in a diverse range of studies with varying methodologies, posing challenges in synthesizing findings. Finally, the study focuses predominantly on immediate impacts and focuses less on long-term effects. Future research should include conflict settings, employ rigorous methodologies, focus on long-term consequences, and foster interdisciplinary collaboration to develop comprehensive crisis intervention strategies.

## Supporting information

**S1 File. PRISMA-ScR-Fillable-Checklist.**
(DOCX)

**S2 File. Selected documents for in-depth analysis.**
(DOCX)

## Author contributions

**Conceptualization:** Xiaoshan Hu.

**Data curation:** Xiaoshan Hu, Jihua Liu, Bingyu Hao, Yang Lv.

**Formal analysis:** Xiaoshan Hu, Jihua Liu, Bingyu Hao, Yang Lv.

**Funding acquisition:** Xiaoshan Hu.

**Investigation:** Jihua Liu, Bingyu Hao, Yang Lv.

**Methodology:** Xiaoshan Hu, Jihua Liu, Bingyu Hao, Yang Lv.

**Project administration:** Jihua Liu, Bingyu Hao, Yang Lv.

**Resources:** Xiaoshan Hu, Jihua Liu, Yang Lv.

**Software:** Xiaoshan Hu, Jihua Liu, Yang Lv.

**Supervision:** Jihua Liu, Bingyu Hao, Yang Lv.

**Validation:** Jihua Liu, Bingyu Hao, Yang Lv.

**Visualization:** Xiaoshan Hu, Yang Lv.

**Writing – original draft:** Xiaoshan Hu, Jihua Liu, Bingyu Hao, Yang Lv.

**Writing – review & editing:** Xiaoshan Hu, Jihua Liu, Bingyu Hao, Yang Lv.

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
