## [Decision Letter · Decision Letter 0]

25 Jul 2024

PONE-D-24-11145Impact of crisis intervention on mental health in the context of emergencies: A scoping reviewPLOS ONE

Dear Dr. Lv,

Thank you for submitting your manuscript to PLOS ONE. After careful consideration, we feel that it has merit but does not fully meet PLOS ONE’s publication criteria as it currently stands. Therefore, we invite you to submit a revised version of the manuscript that addresses the points raised during the review process.

I would like you to go thoroughly through the reviews. Namely, there is a review asking for major revision thus I believe that you have to answer all the comments you get from each reviewer and to improve, based on their observations, you paper which I consider very interesting. This is required for paper to be accepted.As well, I would strongly recommend to check the language and grammar through the whole paper. Additionally, I recommend you to go through your paper and check if all the requirements are met for scoping review.

We look forward to receiving your revised manuscript.

Kind regards,

Iskra Alexandra Nola

Academic Editor

PLOS ONE

Journal Requirements:

"This study is funded by the Sichuan Compulsory Education High-Quality Development Research Center (Project Number: YWYB-2023-03), the Sichuan Primary and Secondary School Teachers' Professional Development Research Center (Project Number: PDTR2022-34), and Center for Education Research at Sichuan Province (Project Number: TER2022-012)."

3. In the online submission form, you indicated that "Data will be available upon request."

Reviewers' comments:

Reviewer's Responses to Questions

**Comments to the Author**

1. Is the manuscript technically sound, and do the data support the conclusions?

Reviewer #1: Yes

Reviewer #2: Yes

Reviewer #3: Yes

2. Has the statistical analysis been performed appropriately and rigorously? 

Reviewer #1: Yes

Reviewer #2: N/A

Reviewer #3: N/A

3. Have the authors made all data underlying the findings in their manuscript fully available?

Reviewer #1: Yes

Reviewer #2: No

Reviewer #3: Yes

4. Is the manuscript presented in an intelligible fashion and written in standard English?

Reviewer #1: No

Reviewer #2: Yes

Reviewer #3: No

5. Review Comments to the Author

Reviewer #1: ethics involves the application of fundamental ethical principles to research activities which include the design and implementation of research, society and others, the use of resources and research outputs, scientific misconduct and the regulation of research.Given the importance of ethics for the conduct of research, it should come as no surprise that many different professional associations, government agencies, and universities have adopted specific codes, rules, and policies relating to research ethics. Many government agencies have ethics rules for funded researchers

Reviewer #2: Thank you to the authors for this very important and enriching scoping review. I suggest the following revisions before publication:

Introduction: The importance of this topic would be even better if you present some quantitative data, e.g. Charlson et al (2017) in Lancet found that 22% of people affected by conflict suffer from psychological disorders.

Methodology:

- Key words could have been broader (psychosocial interventions, psychosocial consequences, acute crisis, mental well-being, psychosocial impact, traumatic event..)

- can you explain why conflict settings have been excluded, as they are a humanitarian crisis? They are not mentioned anywhere

- Align the research questions/focus that you mention in the Abstract, Methodology, Research Protocol etc. They seem to be slightly different.

Results:

o Numeration is wrong

o Table 1: What do you mean with research start time given in months?; include the population

o Table 3: revise table, unclear what are the numbers behind anxiety e.g. Is pressure an official psychological term?

o 2.4 The relationship between crisis intervention and mental health: The reasoning of the interpretation of the spiderweb is difficult to follow. I suggest a quick presentation of the results before the interpretation; how have the themes been chosen? Also, there are some words, such as article, literature, participant that I do not see as key words and could be excluded.

o For the chapters 2.5.1 Direct impact and 2.5.2 indirect impact, there are only 2, respective 1 article that are being discussed that seems to be little.

In the annexe, I would suggest one table with all the articles to quickly be able to jump to the articles with main features

Be careful with terminology “this study provides evidence...” which is not the goal of a scoping review; or “direct effect on mental health...” that is only applicable if it was a RCT that controlled for other variables.

Wishing you good continuation of your research, this is what is needed!

Reviewer #3: 1. The manuscript is technically sound, the rationale, the methodology, presentation of results, discussions and conclusions are coherent. The methodology was rigorous, and conclusions were made based on results. All the research inquiries were answered by the results of the study.

2. Narrative synthesis was used to simply put the different findings together. It is suitable for scoping studies due to its adaptability in accommodating different study designs. Statistical analysis is not applicable for the study design

3. Data can be made available upon request.

4. There manuscript has some clumsy sentences and some grammar errors to be corrected.

6. PLOS authors have the option to publish the peer review history of their article (what does this mean? ). If published, this will include your full peer review and any attached files.

**Do you want your identity to be public for this peer review?** For information about this choice, including consent withdrawal, please see our Privacy Policy .

Reviewer #1: **Yes: ** Nigar Arif-Poladlı

Reviewer #2: No

Reviewer #3: No

---

## [Author Response · Author response to Decision Letter 1]

31 Jul 2024

Dear Honorable Editor and Reviewers,

We would like to show our sincere gratitude to the honorable editor and respected reviewers for giving us a chance to resubmit the manuscript. We believe that the revised version has fulfilled all the lacking of the manuscript and improved its quality. We have used blue color to show the correction marks throughout the manuscript.

Reviewer #1’s comments

General comments

1. Is the manuscript technically sound, and do the data support the conclusions?

Yes.

2. Has the statistical analysis been performed appropriately and rigorously?

Yes.

3. Have the authors made all data underlying the findings in their manuscript fully available?

No.

4. Is the manuscript presented in an intelligible fashion and written in standard English?

Yes.

Author’s reply: Many thanks to respected reviewer for kind observation. We have addressed all issues in the respective section point-by-point.

Specific comments

Reviewer: Ethics involves the application of fundamental ethical principles to research activities which include the design and implementation of research, society and others, the use of resources and research outputs, scientific misconduct and the regulation of research. Given the importance of ethics for the conduct of research, it should come as no surprise that many different professional associations, government agencies, and universities have adopted specific codes, rules, and policies relating to research ethics. Many government agencies have ethics rules for funded researchers.

Author’s reply: Thank you so much for the detailed review of the manuscript. We have followed the ethical guidelines throughout our research.

We are grateful to you for your kind suggestions, time and input for us. We belief your constructive suggestion helps us a lot. Without your kind observation, it would not be possible to improve the manuscript’s quality.

Reviewer #2’s comments

General comments

1. Is the manuscript technically sound, and do the data support the conclusions?

Yes.

2. Has the statistical analysis been performed appropriately and rigorously?

N/A.

3. Have the authors made all data underlying the findings in their manuscript fully available?

No.

4. Is the manuscript presented in an intelligible fashion and written in standard English?

Yes.

Author’s reply: Many thanks to respected reviewer for kind observation. We have addressed all issues in the respective section point-by-point.

Specific comments

Reviewer: Thank you to the authors for this very important and enriching scoping review. I suggest the following revisions before publication.

Author’s reply: Thank you so much for the detailed review of the manuscript. We have addressed all the suggestions point-by-point and responded to all comments below. The line numbers show the corrections of the revised manuscript.

Reviewer: 1. Introduction: The importance of this topic would be even better if you present some quantitative data, e.g. Charlson et al (2017) in Lancet found that 22% of people affected by conflict suffer from psychological disorders.

Author’s reply: Thank you so much for your suggestion. We have added some statistics in the first paragraph of the introduction section according to your kind suggestion.

Reviewer: 2. Methodology:

- Key words could have been broader (psychosocial interventions, psychosocial consequences, acute crisis, mental well-being, psychosocial impact, traumatic event..)

- can you explain why conflict settings have been excluded, as they are a humanitarian crisis? They are not mentioned anywhere

- Align the research questions/focus that you mention in the Abstract, Methodology, Research Protocol etc. They seem to be slightly different.

Author’s reply: We appreciate your critical observation.

-We have revised our keywords section of the research protocol to align with our search string.

-We have added a limitation regarding the exclusion of conflict settings in our conclusion section. Conflict settings have been excluded to maintain a specific focus on natural hazards and public health emergencies, ensuring a clear analysis of intervention strategies pertinent to these contexts. Future research could separately address conflict settings to provide a detailed understanding of crisis interventions in those contexts. Please see the conclusion section.

-We have revised our research question section of the research protocol to align the research question with abstract (overall research problem), introduction and methodology section.

Reviewer: 3. Results:

o Numeration is wrong

o Table 1: What do you mean with research start time given in months?; include the population

o Table 3: revise table, unclear what are the numbers behind anxiety e.g. Is pressure an official psychological term?

o 2.4 The relationship between crisis intervention and mental health: The reasoning of the interpretation of the spiderweb is difficult to follow. I suggest a quick presentation of the results before the interpretation; how have the themes been chosen? Also, there are some words, such as article, literature, participant that I do not see as key words and could be excluded.

o For the chapters 2.5.1 Direct impact and 2.5.2 indirect impact, there are only 2, respective 1 article that are being discussed that seems to be little.

Author’s reply: Many thanks for your critical observations and suggestions.

-We have corected numeration. Please see the results section.

- "Research Start Time" should be "Intervention Time". The "Intervention Time" given in months indicates the period during which the studies began their intervention. For example, "Intervention 0-Jan" refers to studies that started interventions immediately or within the first month. Please see the revised Table 2.

- We appreciate your critical observation. We have revised the mentioned table. “Pressure” should be ‘Stress”. We are sorry for our typos. Please see the revise Table 4.

- We have generated the Figure 2 again using VOS viewer, and added it. We have also updated the explanation.

Reviewer: 4. In the annexe, I would suggest one table with all the articles to quickly be able to jump to the articles with main features.

Author’s reply: Thanks for your kind suggestions. We have added the list of the selected documents as “Supplementary 2. Selected documents for in-depth analysis”.

Reviewer: 5. Be careful with terminology “this study provides evidence...” which is not the goal of a scoping review; or “direct effect on mental health...” that is only applicable if it was a RCT that controlled for other variables.

Author’s reply: We appreciate your critical observations, and suggestions. We have checked the whole manuscript and corrected the wording.

Reviewer: 6 Wishing you good continuation of your research, this is what is needed!.

Author’s reply: Many thanks for your suggestions. We have revised and updated the conclusion section by incorporating your all suggestions.

We are grateful to you for your kind suggestions, time and input for us. We belief your constructive suggestion helps us a lot. Without your kind observation, it would not be possible to improve the manuscript’s quality.

Reviewer #3’s comments

General comments

1. Is the manuscript technically sound, and do the data support the conclusions?

Yes.

2. Has the statistical analysis been performed appropriately and rigorously?

N/A.

3. Have the authors made all data underlying the findings in their manuscript fully available?

Yes.

4. Is the manuscript presented in an intelligible fashion and written in standard English?

No.

Author’s reply: Many thanks to respected reviewer for kind observation. We have addressed all issues in the respective section point-by-point.

Specific comments

Reviewer: 1. The manuscript is technically sound, the rationale, the methodology, presentation of results, discussions and conclusions are coherent. The methodology was rigorous, and conclusions were made based on results. All the research inquiries were answered by the results of the study.

Author’s reply: Thank you so much for the detailed review of the manuscript. We have addressed all the suggestions point-by-point and responded to all comments below. We have used blue color to mark the corrections. The line numbers show the corrections of the revised manuscript.

Reviewer: 1. The authors are recommended to include the keywords used for document selection and the major findings of the key research in the relevant section.

Author’s reply: We appreciate your kind observation and comment. We have added the keywords used for document selection in our research protocol Table. Please see Table 1. Besides, we have also added the keyword network analysis in the section 3.4 with Figure 2 under results section.

Reviewer: 2. Narrative synthesis was used to simply put the different findings together. It is suitable for scoping studies due to its adaptability in accommodating different study designs. Statistical analysis is not applicable for the study design.

Author’s reply: We appreciate your critical observation. In our study, we focused on social, economic and environmental vulnerability, and discussed in detail. Please see sub-section 4.2.2.

Reviewer: 3. Data can be made available upon request.

Author’s reply: Many thanks for suggestion. We have added the list of the selected documents as “Supplementary 2. Selected documents for in-depth analysis”.

Reviewer: 4. There manuscript has some clumsy sentences and some grammar errors to be corrected.

Author’s reply: Thanks for your kind suggestions. We have checked the whole manuscript and corrected the clumsy sentences and grammar errors. Please see the highlighted texts.

We are grateful to you for your kind suggestions, time and input for us. We belief your constructive suggestion helps us a lot. Without your kind observation, it would not be possible to improve the manuscript’s quality.

---

## [Decision Letter · Decision Letter 1]

31 Oct 2024

PONE-D-24-11145R1Impact of crisis intervention on mental health in the context of emergencies: A scoping reviewPLOS ONE

Dear Dr. Lv,

Thank you for submitting your manuscript to PLOS ONE. After careful consideration, we feel that it has merit but does not fully meet PLOS ONE’s publication criteria as it currently stands. Therefore, we invite you to submit a revised version of the manuscript that addresses the points raised during the review process.

We look forward to receiving your revised manuscript.

Kind regards,

Yu Xiao

Academic Editor

PLOS ONE

Journal Requirements:

Reviewers' comments:

Reviewer's Responses to Questions

**Comments to the Author**

1. If the authors have adequately addressed your comments raised in a previous round of review and you feel that this manuscript is now acceptable for publication, you may indicate that here to bypass the “Comments to the Author” section, enter your conflict of interest statement in the “Confidential to Editor” section, and submit your "Accept" recommendation.

Reviewer #1: (No Response)

Reviewer #3: All comments have been addressed

Reviewer #4: (No Response)

2. Is the manuscript technically sound, and do the data support the conclusions?

Reviewer #1: Yes

Reviewer #3: Yes

Reviewer #4: Partly

3. Has the statistical analysis been performed appropriately and rigorously? 

Reviewer #1: No

Reviewer #3: N/A

Reviewer #4: N/A

4. Have the authors made all data underlying the findings in their manuscript fully available?

Reviewer #1: Yes

Reviewer #3: Yes

Reviewer #4: Yes

5. Is the manuscript presented in an intelligible fashion and written in standard English?

Reviewer #1: Yes

Reviewer #3: Yes

Reviewer #4: Yes

6. Review Comments to the Author

Reviewer #1: (No Response)

Reviewer #3: Some of the clumpsy sentences are clupmsy. There is an inconsistency in the use of terms, for instance in the introduction, the use of crisis treatments in the place of crisis intervention.

Reviewer #4: This title of this paper is "Impact of crisis intervention on mental health in the context of emergencies". My strong recommendation is to change the title to say "Impact of crisis intervention on mental health in the context of specific civilian emergencies" The paper leaves out a huge literature, not only on conflict zones, but also on the impact of genocide, terrorist attacks, etc. About one third of the articles focus on COVID, and most of the other articles look at clinical emergencies and responses by law enforcement and health care workers. The introduction to the article should make this clear from the outset. To elaborate further, there is a huge literature on terrorist attacks (such as 9/11), genocides (such as in Rwanda), the atrocities of war (right now in Gaza, Ukraine and Sudan, to give a few examples) and the overlapping profound consequences of rape. The authors barely touch on this very vast literature. Therefore, throughout the paper, the authors need make their focus on specific civilian events clear and avoid references to the many topics that they barely cover.

7. PLOS authors have the option to publish the peer review history of their article (what does this mean? ). If published, this will include your full peer review and any attached files.

**Do you want your identity to be public for this peer review?** For information about this choice, including consent withdrawal, please see our Privacy Policy .

Reviewer #1: **Yes: ** Nigar Arif-Poladlı

Reviewer #3: No

Reviewer #4: **Yes: ** Francine Cournos

---

## [Author Response · Author response to Decision Letter 2]

13 Nov 2024

Dear Honorable Editor and Reviewers,

We would like to show our sincere gratitude to the honorable editor and respected reviewers for giving us a chance to resubmit the manuscript. We believe that the revised version has fulfilled all the lacking of the manuscript and improved its quality. We have used blue color to show the correction marks throughout the manuscript.

Journal Requirements:

Author’s reply: Thank you for your valuable feedback on our manuscript. We have carefully reviewed the reference list to ensure that it is complete, accurate, and up-to-date. Additionally, we did not cite any retracted papers in our manuscript.

Reviewer #1’s comments

1. If the authors have adequately addressed your comments raised in a previous round of review and you feel that this manuscript is now acceptable for publication, you may indicate that here to bypass the “Comments to the Author” section, enter your conflict of interest statement in the “Confidential to Editor” section, and submit your "Accept" recommendation.

No Response

2. Is the manuscript technically sound, and do the data support the conclusions?

Yes.

3. Has the statistical analysis been performed appropriately and rigorously?

Yes.

4. Have the authors made all data underlying the findings in their manuscript fully available?

No.

5. Is the manuscript presented in an intelligible fashion and written in standard English?

Yes.

6. Review Comments to the Author

Reviewer #1: No Response

Author’s reply: Many thanks to respected reviewer for kind observation. We have addressed all issues in the respective section point-by-point.

Reviewer #2’s comments

General comments

Reviewer #1’s comments

1. If the authors have adequately addressed your comments raised in a previous round of review and you feel that this manuscript is now acceptable for publication, you may indicate that here to bypass the “Comments to the Author” section, enter your conflict of interest statement in the “Confidential to Editor” section, and submit your "Accept" recommendation.

Response: All comments have been addressed

Author’s reply: Many thanks to respected reviewer for kind observation, and acceptance. We have addressed all issues in the respective section point-by-point.

2. Is the manuscript technically sound, and do the data support the conclusions?

Yes.

Author’s reply: We appreciate your kind observation.

3. Has the statistical analysis been performed appropriately and rigorously?

N/A.

Author’s reply: Many thanks.

4. Have the authors made all data underlying the findings in their manuscript fully available?

Yes.

Author’s reply: We appreciate your kind observation.

5. Is the manuscript presented in an intelligible fashion and written in standard English?

Yes.

Author’s reply: We appreciate your kind observation.

6. Review Comments to the Author

Reviewer #3: Some of the clumpsy sentences are clupmsy. There is an inconsistency in the use of terms, for instance in the introduction, the use of crisis treatments in the place of crisis intervention.

Author’s reply: Many thanks to respected reviewer for kind observation. We have checked the whole manuscript and corrected the clumpsy sentences. Please see the blue marked sentences of the whole manuscript.

Reviewer #4’s comments

General comments

1. If the authors have adequately addressed your comments raised in a previous round of review and you feel that this manuscript is now acceptable for publication, you may indicate that here to bypass the “Comments to the Author” section, enter your conflict of interest statement in the “Confidential to Editor” section, and submit your "Accept" recommendation.

No Response

2. Is the manuscript technically sound, and do the data support the conclusions?

Partly.

Author’s reply: We appreciate your kind observation. We have addressed your kind concern in the revised manuscript.

3. Has the statistical analysis been performed appropriately and rigorously?

N/A.

Author’s reply: Many thanks.

4. Have the authors made all data underlying the findings in their manuscript fully available?

Yes.

Author’s reply: We appreciate your kind observation.

5. Is the manuscript presented in an intelligible fashion and written in standard English?

Yes.

Author’s reply: We appreciate your kind observation.

6. Review Comments to the Author

Reviewer #4: This title of this paper is "Impact of crisis intervention on mental health in the context of emergencies". My strong recommendation is to change the title to say "Impact of crisis intervention on mental health in the context of specific civilian emergencies" The paper leaves out a huge literature, not only on conflict zones, but also on the impact of genocide, terrorist attacks, etc. About one third of the articles focus on COVID, and most of the other articles look at clinical emergencies and responses by law enforcement and health care workers. The introduction to the article should make this clear from the outset. To elaborate further, there is a huge literature on terrorist attacks (such as 9/11), genocides (such as in Rwanda), the atrocities of war (right now in Gaza, Ukraine and Sudan, to give a few examples) and the overlapping profound consequences of rape. The authors barely touch on this very vast literature. Therefore, throughout the paper, the authors need make their focus on specific civilian events clear and avoid references to the many topics that they barely cover.

Author’s reply: Many thanks to respected reviewer for kind observation. We have revised the title to "Impact of Crisis Intervention on Mental Health in the Context of Specific Civilian Emergencies" to clearly reflect the study’s focus on civilian emergencies not associated with armed conflict, such as natural disasters and public health crises. Additionally, we have incorporated a clarifying statement in the Introduction that acknowledges the extensive research on mental health impacts in conflict settings, including terrorist attacks, genocide, and war-related trauma, while specifying that our study focuses on non-conflict, civilian emergencies. Please see the revised title, and other revision in the second paragraph of the introduction (lines 51-66, and 182-184), methodology (lines 190-193, 281-283, and table 1), results (lines 287-291, 371-376, 412-415, 433-434), discussion (lines 453-460, 467-468, and 603-605), and conclusion (lines 611-612) sections.

We are grateful to you for your kind suggestions, time and input for us. We belief your constructive suggestion helps us a lot. Without your kind observation, it would not be possible to improve the manuscript’s quality.

---

## [Decision Letter · Decision Letter 2]

14 Apr 2025

PONE-D-24-11145R2Impact of crisis intervention on mental health in the context of specific civilian emergenciesPLOS ONE

Dear Dr. Lv,

Thank you for submitting your manuscript to PLOS ONE. After careful consideration, we feel that it has merit but does not fully meet PLOS ONE’s publication criteria as it currently stands. Therefore, we invite you to submit a revised version of the manuscript that addresses the points raised during the review process.

We look forward to receiving your revised manuscript.

Kind regards,

Yu Xiao

Academic Editor

PLOS ONE

Journal Requirements:

Reviewers' comments:

Reviewer's Responses to Questions

**Comments to the Author**

1. If the authors have adequately addressed your comments raised in a previous round of review and you feel that this manuscript is now acceptable for publication, you may indicate that here to bypass the “Comments to the Author” section, enter your conflict of interest statement in the “Confidential to Editor” section, and submit your "Accept" recommendation.

Reviewer #3: (No Response)

Reviewer #5: (No Response)

2. Is the manuscript technically sound, and do the data support the conclusions?

Reviewer #3: Yes

Reviewer #5: (No Response)

3. Has the statistical analysis been performed appropriately and rigorously? 

Reviewer #3: N/A

Reviewer #5: (No Response)

4. Have the authors made all data underlying the findings in their manuscript fully available?

Reviewer #3: Yes

Reviewer #5: (No Response)

5. Is the manuscript presented in an intelligible fashion and written in standard English?

Reviewer #3: Yes

Reviewer #5: Yes

6. Review Comments to the Author

Reviewer #3: 1. Most of the comments from the first review were addressed by the author.

2. The manuscript is technically sound and the data supports the conclusions.

3. The extracted data was analysed and presneted appropriately

4.

52: The use of it's (it is) is not proper for a formal document line a research paper.

53,54,55: The author should note that a public Health emergency could be caused by an armed conflict due to the massive displacement of populations. It is thus important not to use the term to refer only to emergencies of infectious origin like COVID-19. The war in Ukraine and Gaza have resulted in public health emergencies in both settings. Disease outbreak (epidemic) is a better term to use.

51-65 has repetitive information about the focus of the study on emergencies that are not related to armed conflict. It could be presented in a less repititive and concise manner.

68-79: The sentence is long, and clumpsy. Make it shorter and clearer or use a conjunction and a present continous tense

412:Grammar

455-456: I would suggest you parahrase "The analysis focuses on public health interventions within civilian contexts, juxtaposing them with a selection of non-public health-related actions".

498: Repitition of demographic

596, 605: Grammar, assessments and studies

Reviewer #5: The author has revised well on the suggested and commented points. The authors have done their task. Good Job.

7. PLOS authors have the option to publish the peer review history of their article (what does this mean? ). If published, this will include your full peer review and any attached files.

**Do you want your identity to be public for this peer review?** For information about this choice, including consent withdrawal, please see our Privacy Policy .

Reviewer #3: **Yes: ** CHANDINI ALIYOU MOUSTAPHA

Reviewer #5: **Yes: ** Dr. Moses Glorino Rumambo Pandin

---

## [Author Response · Author response to Decision Letter 3]

24 Apr 2025

Dear Honorable Editor and Reviewers,

We would like to express our sincere gratitude to the honorable editor and respected reviewers for the opportunity to resubmit our manuscript. We believe that the revised version has addressed all the identified gaps and significantly improved the overall quality. All corrections have been highlighted in blue throughout the manuscript for ease of review.

Journal Requirements:

Author’s reply: Thank you for your valuable feedback on our manuscript. We have carefully reviewed the reference list to ensure that it is complete, accurate, and up-to-date. Additionally, we did not cite any retracted papers in our manuscript.

Reviewer #3’s comments

1. If the authors have adequately addressed your comments raised in a previous round of review and you feel that this manuscript is now acceptable for publication, you may indicate that here to bypass the “Comments to the Author” section, enter your conflict of interest statement in the “Confidential to Editor” section, and submit your "Accept" recommendation.

No Response

2. Is the manuscript technically sound, and do the data support the conclusions?

Yes.

3. Has the statistical analysis been performed appropriately and rigorously?

Yes.

4. Have the authors made all data underlying the findings in their manuscript fully available?

Yes.

5. Is the manuscript presented in an intelligible fashion and written in standard English?

Yes.

Author’s reply: Thank you to the respected reviewer for your kind observation. We have addressed all issues in the respective section point-by-point.

Reviewer #3’s comments

Reviewer: 1. Most of the comments from the first review were addressed by the author.

Author’s reply: Thank you very much for your kind appreciations.

Reviewer: 2. The manuscript is technically sound and the data supports the conclusions.

Author’s reply: We appreciate your kind observations.

Reviewer: 3. The extracted data was analysed and presneted appropriately

Author’s reply: We appreciate your kind observations.

Reviewer: 4. Minor corrections:

52: The use of it's (it is) is not proper for a formal document line a research paper.

53,54,55: The author should note that a public Health emergency could be caused by an armed conflict due to the massive displacement of populations. It is thus important not to use the term to refer only to emergencies of infectious origin like COVID-19. The war in Ukraine and Gaza have resulted in public health emergencies in both settings. Disease outbreak (epidemic) is a better term to use.

51-65 has repetitive information about the focus of the study on emergencies that are not related to armed conflict. It could be presented in a less repititive and concise manner.

68-79: The sentence is long, and clumpsy. Make it shorter and clearer or use a conjunction and a present continous tense

412: Grammar

455-456: I would suggest you parahrase "The analysis focuses on public health interventions within civilian contexts, juxtaposing them with a selection of non-public health-related actions".

498: Repitition of demographic

596, 605: Grammar, assessments, and studies

Author’s reply: Thank you for your constructive feedback. We have carefully revised the manuscript in accordance with your suggestions. This includes correcting grammatical issues, improving sentence clarity, refining the use of formal language, and reducing repetition. We have also clarified the distinction between public health emergencies and epidemics, ensuring accurate terminology and a more concise presentation of the study’s scope. We believe these revisions have strengthened the overall clarity and quality of the manuscript. Please see the mentioned paragraphs.

Reviewer #5’s comments

1. If the authors have adequately addressed your comments raised in a previous round of review and you feel that this manuscript is now acceptable for publication, you may indicate that here to bypass the “Comments to the Author” section, enter your conflict of interest statement in the “Confidential to Editor” section, and submit your "Accept" recommendation.

Response: All comments have been addressed

2. Is the manuscript technically sound, and do the data support the conclusions?

Yes.

Author’s reply: We appreciate your kind observation.

3. Has the statistical analysis been performed appropriately and rigorously?

N/A.

Author’s reply: Many thanks.

4. Have the authors made all data underlying the findings in their manuscript fully available?

Yes.

5. Is the manuscript presented in an intelligible fashion and written in standard English?

Yes.

Author’s reply: Thank you to the respected reviewer for your kind observation. We have addressed all issues in the respective section point-by-point.

General comments

Reviewer: The author has revised well on the suggested and commented points. The authors have done their task. Good Job.

Author’s reply: We appreciate your comments and acceptance.

We are truly grateful for your valuable suggestions, time, and thoughtful input. We believe your constructive feedback has been immensely helpful. Without your observations, improving the quality of the manuscript would not have been possible.

---

## [Decision Letter · Decision Letter 3]

5 Aug 2025

PONE-D-24-11145R3Impact of crisis intervention on mental health in the context of specific civilian emergenciesPLOS ONE

Dear Dr. Lv,

Thank you for submitting your manuscript to PLOS ONE. After careful consideration, we feel that it has merit but does not fully meet PLOS ONE’s publication criteria as it currently stands. Therefore, we invite you to submit a revised version of the manuscript that addresses the points raised during the review process.

We look forward to receiving your revised manuscript.

Kind regards,

Yu Xiao

Academic Editor

PLOS ONE

Journal Requirements:

**Additional Editor Comments:**

Authors, please note that this will be the last round of revision requests. Please address the comments raised by Reviewer 6.

Reviewers' comments:

Reviewer's Responses to Questions

**Comments to the Author**

1. If the authors have adequately addressed your comments raised in a previous round of review and you feel that this manuscript is now acceptable for publication, you may indicate that here to bypass the “Comments to the Author” section, enter your conflict of interest statement in the “Confidential to Editor” section, and submit your "Accept" recommendation.

Reviewer #6: All comments have been addressed

2. Is the manuscript technically sound, and do the data support the conclusions?

Reviewer #6: Yes

3. Has the statistical analysis been performed appropriately and rigorously? 

Reviewer #6: N/A

4. Have the authors made all data underlying the findings in their manuscript fully available?

Reviewer #6: Yes

5. Is the manuscript presented in an intelligible fashion and written in standard English?

Reviewer #6: Yes

6. Review Comments to the Author

Reviewer #6: This is an important, comprehensive and timely review on crisis intervention. The paper has already undergone several rounds of reviews, achieving significant improvements. This is due to the efforts of both the authors and the reviewers!

From that point on, I was asked to participate as a new reviewer. Naturally, new reviewers find new points of criticism that can be improved. These include platitudes such as "Psychological crisis intervention is widely considered one of the most effective approaches for managing mental health crises" (lines 93-94), which are not cited and contribute nothing in terms of content. Or replace “Table 2. Summary of basic characteristics of the literature” with included studies: in PICO format, for quality assessment, or similar. However, in my opinion, the standard of the work in its current form is already sufficient for publication. If there really needs to be any changes to the review, I would shorten the introduction by lines 154-164. The three questions mentioned (160-164) are either not addressed in the review or are only addressed in passing. The central questions (which are significantly different) appear in Chapter 2.2 and are then adequately addressed in the review.

Even though I personally would have approached this review with different priorities, relevant content is presented here that will probably be frequently cited.

7. PLOS authors have the option to publish the peer review history of their article (what does this mean? ). If published, this will include your full peer review and any attached files.

**Do you want your identity to be public for this peer review?** For information about this choice, including consent withdrawal, please see our Privacy Policy .

Reviewer #6: **Yes: ** Ulrich Wesemann

---

## [Author Response · Author response to Decision Letter 4]

9 Aug 2025

Dear Honorable Editor and Reviewers,

We would like to express our sincere gratitude to the honorable editor and respected reviewers for the opportunity to resubmit our manuscript. We believe that the revised version has addressed all the identified gaps and significantly improved the overall quality. All corrections have been highlighted in blue throughout the manuscript for ease of review.

Journal Requirements:

Author’s reply: Thank you very much for your kind suggestion. The reviewers did not ask to cite any articles.

Author’s reply: Thank you for your valuable feedback on our manuscript. We have carefully reviewed the reference list to ensure that it is complete, accurate, and up-to-date. Additionally, we did not cite any retracted papers in our manuscript.

Additional Editor Comments:

Authors, please note that this will be the last round of revision requests. Please address the comments raised by Reviewer 6.

Author’s reply: Thank you very much for your kind suggestions. We have addressed all issues in the respective section point-by-point.

Reviewer #6’s general comments

1. If the authors have adequately addressed your comments raised in a previous round of review and you feel that this manuscript is now acceptable for publication, you may indicate that here to bypass the “Comments to the Author” section, enter your conflict of interest statement in the “Confidential to Editor” section, and submit your "Accept" recommendation.

Reviewer #6: All comments have been addressed

2. Is the manuscript technically sound, and do the data support the conclusions?

Reviewer #6: Yes.

3. Has the statistical analysis been performed appropriately and rigorously?

Reviewer #6: N/A.

4. Have the authors made all data underlying the findings in their manuscript fully available?

Reviewer #6: Yes.

5. Is the manuscript presented in an intelligible fashion and written in standard English?

Reviewer #6: Yes.

Author’s reply: Thank you to the respected reviewer for your kind observation. We have addressed all issues in the respective section point-by-point.

Reviewer #6’s comments

Comment 1. This is an important, comprehensive and timely review on crisis intervention. The paper has already undergone several rounds of reviews, achieving significant improvements. This is due to the efforts of both the authors and the reviewers!

Author’s reply: Thank you very much for your kind appreciations.

Comment 2. From that point on, I was asked to participate as a new reviewer. Naturally, new reviewers find new points of criticism that can be improved. These include platitudes such as "Psychological crisis intervention is widely considered one of the most effective approaches for managing mental health crises" (lines 93-94), which are not cited and contribute nothing in terms of content. Or replace “Table 2. Summary of basic characteristics of the literature” with included studies: in PICO format, for quality assessment, or similar.

Author’s reply: We appreciate the reviewer’s careful reading and constructive feedback. In line with your suggestion, the sentence “Psychological crisis intervention is widely considered one of the most effective approaches for managing mental health crises” (formerly in lines 93–94) has been removed from the manuscript to avoid unsubstantiated generalizations and to maintain a focus on evidence-based statements. We have ensured that similar generic statements are either deleted or supported with relevant citations.

Regarding the suggestion to replace Table 2 with a PICO-format summary, we acknowledge the value of such structuring. However, Table 2 in our manuscript serves to present the basic characteristics of the included literature in alignment with the PRISMA-ScR framework for scoping reviews, which emphasizes descriptive mapping rather than PICO extraction. Since our objective was to provide an overview of study settings, populations, intervention timelines, and designs, we have retained the existing format, which we believe remains consistent with scoping review reporting standards.

Comment 3. However, in my opinion, the standard of the work in its current form is already sufficient for publication. If there really needs to be any changes to the review, I would shorten the introduction by lines 154-164. The three questions mentioned (160-164) are either not addressed in the review or are only addressed in passing. The central questions (which are significantly different) appear in Chapter 2.2 and are then adequately addressed in the review.

Author’s reply: We appreciate your kind observations. We have removed the mentioned sentences and did sub-sequent changes of the relevant contents in the methodology section to address your kind concerns. Please see lines 163-164, and Table 1.

Comment 4. Even though I personally would have approached this review with different priorities, relevant content is presented here that will probably be frequently cited.

Author’s reply: We appreciate your observation. We are truly grateful for your valuable suggestions, time, and thoughtful input. We believe your constructive feedback has been immensely helpful. Without your observations, improving the quality of the manuscript would not have been possible.

---

## [Editor Report · Decision Letter 4]

13 Aug 2025

Impact of crisis intervention on mental health in the context of specific civilian emergencies

PONE-D-24-11145R4

Dear Dr. Lv,

We’re pleased to inform you that your manuscript has been judged scientifically suitable for publication and will be formally accepted for publication once it meets all outstanding technical requirements.

Kind regards,

Yu Xiao

Academic Editor

PLOS ONE

Additional Editor Comments (optional):

The reviewers were basically satisfied with the author's revisions. Congratulations!
---

## [Editor Report · Acceptance letter]

PONE-D-24-11145R4

PLOS ONE

Dear Dr. Lv,

I'm pleased to inform you that your manuscript has been deemed suitable for publication in PLOS ONE. Congratulations! Your manuscript is now being handed over to our production team.

Kind regards,

on behalf of

Dr. Yu Xiao

Academic Editor

PLOS ONE